# Effects of Cannabidiol on Innate Immunity: Experimental Evidence and Clinical Relevance

**DOI:** 10.3390/ijms24043125

**Published:** 2023-02-04

**Authors:** Stefano Martini, Alessandra Gemma, Marco Ferrari, Marco Cosentino, Franca Marino

**Affiliations:** Center for Research in Medical Pharmacology, University of Insubria, 21100 Varese, Italy

**Keywords:** cannabidiol, innate immunity, PMN

## Abstract

Cannabidiol (CBD) is the main non-psychotropic cannabinoid derived from cannabis (*Cannabis sativa* L., fam. Cannabaceae). CBD has received approval by the Food and Drug Administration (FDA) and European Medicines Agency (EMA) for the treatment of seizures associated with Lennox–Gastaut syndrome or Dravet syndrome. However, CBD also has prominent anti-inflammatory and immunomodulatory effects; evidence exists that it could be beneficial in chronic inflammation, and even in acute inflammatory conditions, such as those due to SARS-CoV-2 infection. In this work, we review available evidence concerning CBD’s effects on the modulation of innate immunity. Despite the lack so far of clinical studies, extensive preclinical evidence in different models, including mice, rats, guinea pigs, and even ex vivo experiments on cells from human healthy subjects, shows that CBD exerts a wide range of inhibitory effects by decreasing cytokine production and tissue infiltration, and acting on a variety of other inflammation-related functions in several innate immune cells. Clinical studies are now warranted to establish the therapeutic role of CBD in diseases with a strong inflammatory component, such as multiple sclerosis and other autoimmune diseases, cancer, asthma, and cardiovascular diseases.

## 1. Introduction

In the last two decades, the clinical use of cannabinoids has been increasing [1,2,3]. Cannabinoids include *Cannabis sativa* L. derivatives, more than 100 terpenophenolic secondary metabolites named phytocannabinoids (to distinguish them from synthetic and endocannabinoids), such as the psychoactive Δ9-tetrahydrocannabinol (Δ9-THC), the non-psychotropic cannabidiol (CBD), cannabinol (CBN), cannabigerol (CBG), cannabichromene (CBC), and cannabidivarin (CBDV) [4]. Δ9-THC and CBD are the best known and more used in clinical practice. Both are derived from their acidic precursors Δ9-tetrahydrocannabinolic acid (THCA) and cannabidiolic acid (CBDA), which are then decarboxylated to Δ9-THC or CBD [5]. Different cannabinoid-based drugs have been studied, some of which have been approved in various countries for the treatment of different disorders [6]. A summary of such drugs is listed in Table 1.

It is worth noting that CBD, unlike other cannabinoids, shows no signals of drug abuse liability [7,8] and no significant side-effects at therapeutic doses [9].

Clinical studies suggest a potential effect of CBD in several conditions including anxiety [4], psychiatric disorders [10], and epilepsy [11]. CBD is also used in the treatment of cancer-related nausea and vomiting, of spasms and pain in multiple sclerosis [1,12,13,14,15], and of peripheral neuropathic pain due to diabetic condition [16]. CBD use has also been suggested to treat seizures in Dravet and Lennox–Gastaut patients [17] and for the treatment of psychotic symptoms in Parkinson’s disease [18].

Interestingly, some of the abovementioned clinical effects could also be related to its anti-inflammatory properties [19,20,21]. In this regard, in vitro and ex vivo studies show that CBD exerts its anti-inflammatory action by modulating the release of proinflammatory cytokines such as tumour necrosis factor (TNF)-α, nuclear factor kappa-light-chain-enhancer of activated B cells (NF-κB), or peroxisome proliferator-activated receptor (PPAR), as well as their interaction with transcription factors [19,22,23]. Moreover, CBD has been shown to potentially affect the cyclooxygenase (COX) pathway [24,25,26]. 

Despite all abovementioned data, to the best of our knowledge, no published studies have investigated the possible clinical implications of CBD use in humans and its therapeutic relevance as a possible anti-inflammatory compound. In the present review, we revise the available literature concerning preclinical studies which evaluate CBD’s effects on immuno-inflammatory processes involving cells of the innate immune system. A better understanding of the possible role of CBD in the modulation of inflammatory mechanisms in innate immunity could help in outlining the clinical use of CBD-based compounds in inflammatory pathologies and open the way to new strategies for their clinical use, also in view of the safety and effectiveness which CBD has shown, so far, in the treatment of some pathological conditions.

### 1.1. CBD Pharmacodynamics

CBD’s effects are mainly exerted through action on cannabinoid receptors type 1 (CB1R) and type 2 (CB2R) [27]. CB1 receptors are mainly expressed in the central nervous system (CNS) [28], whereas CB2 ones are mainly present on immune cells such as polymorphonuclear leukocytes (PMNs) and lymphocytes [29,30,31].

Although the functions of these receptors are very complex, stimulated CB1 receptors perform their actions through G α i/o activation, with consequent inhibition of the adenylate cyclase enzyme synthesis, resulting in a decrease in cyclic adenosine monophosphate (cAMP) levels, and an elevated level of mitogen-activated protein kinases (MAPKs) [32]. 

CBD also binds several other targets such as transient receptor potential (TRP) channels, including TRPA (A for ankyrin), TRP M (M for melastatin), and TRPV (V for vanilloid) [33], acting in particular as an agonist of TRPV member 1, which is expressed in neurons and immune cells, such as PMNs, dendritic cells, macrophages, and T lymphocytes [34]. Lastly, CBD may act as an agonist of serotonin receptor member 1a (5-HT1A) [35] and adenosine A2A receptors [36], and possibly as an allosteric modulator of μ and δ opioid receptors [37]. The very complex pharmacodynamic profile of CBD was elegantly described and resumed in a recent review [13]. An updated list of identified CBD targets, with reported effects and binding affinities, is summarised in Table 2.

### 1.2. CBD Pharmacokinetics

Relevant pharmacokinetic data for CBD are summarised in Table 3.

The pharmacokinetic profile of CBD shows that absorption and distribution are influenced by the P-glycoprotein (P-gp) and, possibly, ATP-binding cassette subfamily C member 5 (ABCC5), both of which are efflux pumps [46,47]. CBD is metabolised by the cytochrome P450 (CYP450) superfamily of enzymes, particularly CYP3A4 and CYP2C9 [48,49]. The UDP-glucuronosyltransferase (UGT) enzyme family is also involved in CBD biotransformation (Stout et al., 2014), particularly UGT1A9.

The major circulating metabolite is 7-carboxy-CBD (7-COOH-CBD), followed by parent CBD, 7-hydroxy-CBD (7-OH-CBD; an active metabolite), and 6-hydroxy-CBD (6-OH-CBD; a relatively minor metabolite) [45]. Little is known about the pharmacological activity of CBD metabolites in humans [50].

### 1.3. CBD Pharmacogenetics

To the best of our knowledge, only few studies have investigated the association between the patient’s genetic profile and CBD effects. For example, Davis and colleagues showed, in epilepsy patients, a correlation between single-nucleotide polymorphisms (SNPs, DNA sequence variations occurring when a single nucleotide in the genome differs between paired chromosomes) and response to CBD treatments [47]. The authors found that rs6729738 SNP in the aldehyde oxidase gene and rs12539 SNP in the diamine oxidase gene were more frequent in subjects with low seizure control. The same authors also found that rs1339067 SNP in the *SLC15A1* gene and rs3749442 SNP in the *ABCC5* gene, both associated with a reduction in transporter expression in the CNS, induced lower CBD response and higher side-effects.

However, several other SNPs with biological effects have been identified in genes coding for CBD targets (CB1R, CB2R, and TRPV1) and in key enzymes for CBD metabolism (P-gp, ABCC5, SLC15A1; AOX1, AOC1 CYP2C9, CYP3A4, and UGT1A9); hence, it is not possible to exclude that these SNPs could influence response to CBD treatment in different disease conditions. SNPs with potential effects on CBD actions are reported in Table 4.

## 2. Effects of CBD on Immune Systems

The immune system (IS) is an interactive network of immune cells, humoral factors, lymphoid organs, and other products such as cytokines and interleukins. The IS is conventionally divided into two parts, differing in both swiftness and specificity of their action on host defence: innate immunity and adaptive immunity. Although a tight crosstalk is present, with many interactions between the two branches of the IS, cellular components and function regulation remain different and are well characterised [61].

The study of the effects of cannabinoids on IS modulation is mainly focused on innate rather than adaptive immunity; for this reason, the vast majority of reviews in this field took into account the effects of cannabinoids on specific disease models, as well as specific cell types. More in detail, some papers focused on studying the mechanism of action of cannabinoids, using the whole cannabis plant or single active cannabinoids, on the modulation of specific cell subpopulations involved in innate functions, such as macrophages [62] and NK cells [63]. However, to the best of our knowledge, no review has reported CBD’s effects on specific cell subpopulations involved in innate immunity.

On the other hand, some reviews evaluated the effects of CBD, alone or in combination with other active cannabinoids, on specific inflammatory diseases such as inflammatory bowel diseases [64], multiple sclerosis [65], and fibromyalgia [66], as well as pain management in palliative care [67], whereas other works focused on the effects of cannabinoids on infective diseases, such as acquired immune deficiency syndrome (AIDS) [68] and COVID-19 [69,70]. Lastly, the role of CBD has been reported in several reviews, which described it in well-defined pathologies such as epilepsy [71], fibromyalgia [72] or dermatological disorders [73]. 

The approach of this review is quite different compared with previous published reviews. Indeed, rather than focusing on a single disease or cellular subpopulation, we adopt a more holistic approach, in which we report the effects of CBD alone, taking into account all cell populations in the innate immune system, also including the humoral component, which is still almost completely unexplored but potentially valid for adding knowledge about the role of CBD in the immune response (see Figure 1 and Appendix A).

Lastly, we discuss the possible implications of CBD use in many pathologies with an inflammatory component.

### 2.1. CBD’s Action on the Complement System and Antibacterial Peptides

The complement system is a proteolytic cascade, which recognises foreign pathogens and damaged self-cells [74]. To date, just one paper has reported an evaluation of CBD’s effects on the complement system. In a rat model in which schizophrenia symptoms were induced by ketamine (KET) treatment, the transcriptional profile of some complement system genes was studied after rats were treated with CBD. The authors found that C1qa, C1qc, C2, and C3 mRNA expression levels in the prefrontal cortex were reduced in schizophrenic rats, whereas CBD treatment increased their expression. Moreover, the authors found that CBD treatment improved KET-induced schizophrenia symptoms in the model. Since all these genes code for important proteins of the complement system, CBD’s ability to restore their levels (at least for mRNA expression) could suggest that the improvement in cognitive symptoms could be due to a CBD-mediated modulation of the complement system in the CNS [75].

Antibacterial peptides are cationic peptides with antibiotic and immunomodulating properties, e.g., α- and β-defensins, the cathelicidin family, and the saposin family [76]. To the best of our knowledge, no studies have reported the effect of CBD on the modulation of this component of innate immunity.

### 2.2. Monocytes/Macrophages

Monocytes originate from bone marrow progenitors and enter in the peripheral bloodstream. Inflammatory conditions induce monocyte migration into tissues, where their exposure to growth factors, proinflammatory cytokines, and microbes causes their conversion into macrophages [77,78,79]. After tissue infiltration, macrophages contribute to innate immune responses through antigen presentation, the production of several cytokines, such as IL-1, IL-6, and TNF-α, and the phagocytosis process [77,80].

Several in vitro and in vivo studies have evaluated the effects of CBD modulation on macrophages functions, albeit with often contrasting results.

For example, Huang and colleagues found an anti-inflammatory effect for CBD in RAW 264.7 macrophages, with a significant reduction in IL-1β, TNF-α, and IL-6 production in both macrophage [81,82,83] and monocyte [84,85] cell lines, as well as in human monocytes [86]. The inhibitory effect of CBD on proinflammatory cytokine production/release was confirmed in an experimental autoimmune encephalomyelitis (EAE) model, in which Dopkins and colleagues found CBD treatment to reduce IL-1β production [87]. On the other hand, other authors found a proinflammatory effect for CBD. In this regard, Muthumalage and Rahman found a proinflammatory effect for CBD in a U937 macrophage line, with a significant increase in IL-1 receptor, IL-8, IL-16, and IL-32 production. The proinflammatory effect of CBD on macrophages was confirmed in RAW264.7, in which CBD induced, in a dose-dependent manner, an increase in proinflammatory cytokines such as IL-12 [88,89] and a decrease of anti-inflammatory cytokines such as IL-10 [88].

Moving to reactive oxygen species production, CBD treatment was found to inhibit ROS production in monocyte [90] and macrophage [89] cell lines, as well as in monocytes from healthy subjects [91].

Chemotaxis and tissue infiltration are fundamental processes in the inflammatory cascade; for this reason, CBD’s effects on these processes were studied by several authors. The reported results showed that CBD not only inhibits macrophages chemotaxis, as shown by Sacerdote in an in vitro model [88], but also tissue infiltration, as shown in different animal models of inflammatory diseases, such as liver inflammation [81], multiple sclerosis [87], pulmonary hypertension [92], and colitis [83].

The apoptosis process is known to be involved in the regulation of inflammatory processes, with a prevalent anti-inflammatory outcome. Several authors have reported that CBD treatment increased apoptotic processes in a THP-1 monocyte cells line [90], as well as in freshly isolated monocytes from human healthy subjects [91,93].

The effects of CBD on the autophagy process, a self-degradative process which is crucial for balancing energy sources at critical times in development and in response to nutrient stress [94], was studied by Tomer and colleagues, who found that CBD induced an upregulation of autophagy receptor p62/SQSTM1 [95]. The positive effects of CBD on autophagic processes were confirmed by Yeisley and colleagues, who evaluated the expression of the mammalian target of rapamycin (mTOR), a complex implicated in autophagic processes. The authors found that CBD treatment reduced phosphorylated m-TOR Ser 2448 levels, inducing cell autophagy [85]. Altogether, these results seem to suggest that the ability of CBD to increase autophagy in monocytes/macrophages could increase the chance of these cells avoiding programmed cell death, with an overall proinflammatory effect.

In light of these contrasting results, it is possible to hypothesise that proinflammatory effects, shown by CBD in several in vitro/ex vivo models, could be due to a direct and specific action on macrophages, whereas its anti-inflammatory effects, shown in in vivo models, could instead be due to its action on other cell types, such as astrocytes or PMNs.

### 2.3. Glial Cells (Astrocytes, Microglia, and Oligodendrocytes)

Among glial cells, astrocytes are the most abundant, being involved in several physiological functions, including homeostasis maintenance and supporting neural and immune system functions; however, despite their known protective functions, hyperactivation of astrocytes can lead to brain damage [96].

In an in vitro model of the blood–brain barrier (BBB), Hind and colleagues found that CBD decreased cell damage and reduced cell adhesion molecule 1 (VCAM1) and vascular endothelial growth factor (VEGF) expression through the modulation of astrocyte functions [97]. Moreover, Di Giacomo and colleagues showed that CBD was able to decrease reactive oxygen species (ROS) production by astrocytes [98].

CBD’s effects on astrocyte activation were confirmed in in vivo models of transient [99] and global ischaemia [100]. In these models, CBD reduced glial fibrillary acidic protein (GFAP) expression [100] and myeloperoxidase (MPO) activity, which are both markers of astrocytes activation. In addition, Wu et al. observed, in both ex vivo and in vivo models, that CBD exerted an inhibitory effect on IL-6 production and astrocytic proliferation (Wu et al. 2021).

These results suggest a possible role for CBD in preventing astrocytes activation, suggesting a potential role for CBD as a neuroprotective agent [99,100]. 

Microglial cells represent 10–15% of glial cells and play a role in development, maintaining homeostasis, and healing CNS disorders. Microglial cells actively interact with neurons, astrocytes, and blood vessels, promoting the activation of these cells and, thus, increasing their ability to respond to damage or infections with an overall proinflammatory effect [101].

The involvement of CBD in the modulation of microglial cells has mainly been evaluated in in vitro and ex vivo models. CBD was found to be involved in the decrease in microglial activation [102,103,104] and inhibition of ROS release by these cells [105]. Moreover, some authors found that CBD inhibited inflammatory pathways in microglial cells [87,106,107], preventing their activation by decreasing intracellular Ca^2+^ levels [108] and proinflammatory cytokine production [106,109], as well as increasing apoptosis pathways [110].

CBD’s effect on phagocytosis was evaluated in both in vitro and in vivo models. In particular, CBD treatment was shown to result in an increase in phagocytosis [111] and expression of a receptor involved in the phagocytosis process, such as transient receptor potential cation channel subfamily V (TRPV) member 2 [112].

The mechanism of action via which CBD exerts its effect on microglial cells has been evaluated by Wu and colleagues using specific agonists and antagonists for CB1/2, TRPV1, and GPR55 receptors, without finding any correlation between action on these receptors and CBD-mediated apoptosis in microglial cells. In order to explain these results, the authors postulated the involvement of other known CBD targets, such as TRPV2, 5-HT1A, and PPARγ [110]. This hypothesis was corroborated by results reported by Landucci and co-workers, who demonstrated a consistent inhibition of CBD effects on inflammatory microglia phenotype after coincubation with TRPV1, 5-HT1A, and PPARγ antagonists [104]. Two other papers confirmed the role of TRPV2 in the modulation of microglial cell functions. In particular, in microglial cells with TRPV2 knockdown, the CBD-induced increase in mRNA expression of phagocytosis-related receptors was found to be abolished [111,112].

Oligodendrocytes are cells generated from oligodendrocyte progenitor cells (OPCs). Throughout the CNS, OPCs represent proliferating and migrating progenitor cells, which can transform into oligodendrocytes. The main function of oligodendrocytes is to generate myelin, a lipid membrane which wraps tightly around axons to allow for rapid nerve conduction [113]. The effects of CBD on oligodendrocytes and OPC is still unclear. Indeed, on the one hand, CBD’s effects on resting oligodendrocytes include an increase in ROS generation, which suggests a proinflammatory effect for CBD; on the other hand, the activation of apoptotic processes in these cells supports an anti-inflammatory role for CBD [114,115].

Mecha and colleagues also evaluated the mechanism of action via which CBD acts on OPC functions through the coincubation of CBD and antagonists for CB1/2, TRPV1, and PPARγ receptors. Since the authors found that CBD antagonists do not reverse CBD’s effects on ROS production and endoplasmic reticulum stress in OPC, they suggest that CBD’s effects could be mediated by different CBD targets, such as TRPV2, 5-HT1A, PPARγ [115].

### 2.4. Mast Cells

Mast cells are involved in allergy, particularly in the early and acute phases of allergic reactions. These cells exhibit distinct morphological characteristics, including prominent cytoplasmic granules, and produce a variety of mediators, including heparin, histamine, and neutral proteases [116].

Cannabinoids are known to be involved in the modulation of mast cell functions [117]; however, only few studies have reported CBD’s effects on the modulation of these cells. Among these, Giudice et al., using a model of rat basophilic leukaemia mast cell line (RBL-2H3), demonstrated that CBD enhanced the release of β-hexosaminidase, a marker of mast cell activation [118]. On the contrary, in a murine model of intestinal inflammation, CBD inhibited the release of enzymes involved in inflammation and remodelling of the extracellular matrix such as mast cell chymase and metalloproteinase (MMP) 9, both produced and released by mast cells, thus suggesting a possible inhibitory effect of CBD in some key functions of these cells [119].

### 2.5. NK Cells

NK cells represent 5–10% of mononuclear cells in the bloodstream and act as natural cytotoxic cells against tumours, by inhibiting proliferation, migration, and colonisation of distant tissues by metastases [120]. These cells produce a large number of cytokines, including IFN-γ, which modulate adaptive immune responses [121,122]. Recently, NK cells were recognised for their cytotoxicity against normal immune cells, playing an important physiological role in the control of immune responses and homeostasis maintenance [123].

NKs highly express CB2 receptors [124]; at least in these cells, CBR2 activation was found to be crucial for the release of inflammatory cytokines, suggesting that CBR2 and its ligands could play a key inhibitory role for these cell subpopulations [125].

### 2.6. Dendritic Cells

Dendritic cells (DCs), derived from bone marrow stem cells, are the strongest antigen-presenting cells (APCs) of the immune system. They play a crucial role in the triggering of primary immune responses and in the enhancement of secondary immune responses. DCs express CB1 and CB2 receptors [126], and endogenous and exogenous cannabinoids (THC and anandamide) can suppress the immune response, e.g., through their ability to induce apoptosis in DCs [127]. Despite the potential effect that CBDs could exert on immune functions through the modulation of DCs, to the best of our knowledge, the effect of CBD on these cells is still to be investigated.

### 2.7. Eosinophils and Basophils

Eosinophilic cells are typical of all vertebrates, including reptiles, amphibians, mammals, and fish, representing 1% to 3% of circulating leukocytes in mammals [128]. Eosinophils are involved in asthma processes [129], exerting a protective role against parasites such as helminths [130]. Eosinophils are also involved in various allergic and immune-mediated conditions, suggesting an important role in the propagation and potentiation of allergic-type processes within the host [131]. They are part of a complex regulatory network, modulating local and systemic immune and inflammatory responses, together with other immune cell types, including PMNs, lymphocytes, and macrophages [132].

Despite being the smallest population of leukocytes (1%), basophils are crucial in protecting the host against infections through the production of important factors involved in the crosstalk between innate and adaptive immunity. Moreover, basophils produce and release interleukin (IL)-4 and IL-13, which are important for Th2 activation and response [133]. Even though the endocannabinoid system appears to modulate eosinophil and basophil functions [134,135], to the best of our knowledge, no reported data are present in the literature about the ability of CBD to modulate the functions of these cells.

### 2.8. Polymorphonuclear Neutrophils (PMNs)

PMNs, generated in the bone marrow from myeloid precursors, are the most abundant granulocytes and the first immune cells recruited into inflammatory sites [136]. They undergo a process called “priming”, in which exposure to TNF-α, platelet-activating factor (PAF), IFN-γ, G-CSF, and several other interleukins induces a wide range of phenotypic changes, including reduced apoptosis, changes in the expression of receptor and adhesion molecules, phagocytosis, degranulation, and production of proinflammatory mediators [137,138,139]. 

PMNs are the major agents in acute inflammation, but several lines of evidence suggest that they contribute to chronic inflammation as well, e.g., in atherosclerosis [140,141]. Moreover, the involvement of PMNs in different inflammatory pathologies has been extensively studied in diseases with an inflammatory component, such as multiple sclerosis [142,143], neuromyelitis optica [144], inflammatory bowel diseases [145], myositides [146], and cardiovascular diseases [147,148,149,150,151,152].

In this context, CBD, thanks to its anti-inflammatory activity, appears to be a promising potential compound for the treatment of these and other diseases with an important dysregulation in inflammatory processes. In this section, we extensively review the main results reported in the scientific literature regarding the ability of CBD to modulate PMNs functions in both in vitro and in vivo models.

As mentioned above, PMNs are the first cells recruited into inflamed sites, through a multiple-step process, also known as the extravasation cascade.

CBD’s ability to modulate PMN extravasation has been examined in several in vitro models of migration. For example, McHugh and colleagues found that CBD inhibited fMLP-induced migration of human PMNs in a concentration-dependent manner [153], and the results of this study were also confirmed in a similar model by Gómez and colleagues [154], Mukhopadhyay et al. [155], and Mabou Tagne et al., who showed that CBD inhibits PMN migration after activation with IL-8, but not in resting cells, in a concentration-dependent manner [156]. This finding is, in our opinion, of extreme interest, because it seems to suggest that CBD’s anti-inflammatory effects are greater in pathological conditions, whereas this compound does not seem to exert an inhibitory effect on the immune system in physiological conditions, thus once again highlighting the safety and manageability of CBD.

After adhesion, PMNs infiltrate tissues, and this is a key step in the inflammatory process. CBD was found to cause a significant decrease in myeloperoxidase (MPO) activity, a marker for indirect assessment of PMN tissue infiltration [155,157]. 

CBD’s effects on tissues infiltration by PMN has also been studied in different animal models of inflammatory diseases, such as ischaemic brain damage [94], corneal hyperalgesia [158], liver steatosis [159], segmental hepatic ischaemia [155], and periodontitis [157]. Although CBD showed inhibitory effects in all these models, conflicting results were found when tissue infiltration was evaluated in animal models of lipopolysaccharide (LPS)-induced lung injury. In some mouse and guinea pig models, CBD significantly reduced LPS-induced PMN infiltration, and its effect was maintained for several days even after CBD treatment discontinuation [36,160]. These results seem to confirm the anti-inflammatory role of CBD and suggest its usefulness in inflammatory diseases, which require chronic treatment. On the other hand, this result was not confirmed by Karmaus et al., who found an enhancement in LPS-induced PMN accumulation in the lungs of mice treated with CBD, thus suggesting an amplification of inflammatory processes by the compound [161]. Lastly, Makwana and colleagues reported that exposure of guinea pigs to CBD did not reduce PMN recruitment in the airways, which was induced by LPS or TNF-α [162].

ROS production by PMNs is a fundamental function for the proinflammatory action of these cells, including the production of neutrophil extracellular traps (NETs).

CBD inhibited ROS generation in ex vivo models based on PMN from both mice [163] and healthy human subjects [156,159]. The observed effects of CBD on ROS production in ex vivo models were confirmed in animal models, in which ROS production was found to be significantly reduced in PMNs obtained from mice treated with CBD, compared with nontreated mice [159]. Moreover, Biernacki and colleagues evaluated the effects of CBD on ROS production in nude rats irradiated with ultraviolet light, showing CBD to reduce irradiation-induced ROS production [164].

The possibility of slowing stimulus-induced ROS production without affecting resting ROS generation (which is necessary for physiological cell homeostasis) is considered pivotal in stopping the inflammatory cascade; ROS are key mediators involved in several inflammatory diseases, such as cardiovascular diseases and many others [151].

As mentioned above, ROS generation is also involved in NET production. Wójcik et al. indirectly evaluated ROS production through the measurement of NET production in PMNs from psoriasis patients, in which CBD was able to reduce NETotic effects. The ability of CBD to reduce NETosis in PMNs from these subjects suggests a promising effect in decreasing ROS-related oxidative stress, limiting both tissue inflammation and damage in psoriasis and other chronic inflammatory diseases [139].

Lastly, Gómez found that CBD triggered a significant decrease in oxygen uptake and H_2_O_2_ generation but also increased ROS production. In order to explain this discrepancy, the authors postulated that CBD might activate different receptors present on PMNs, which in turn modulate different intracellular pathways, causing these apparently contradictory effects [154]. 

The effects of CBD on cytokine production by PMN have been evaluated in both ex vivo and in vivo models. In particular, in ex vivo models, PMN activation was shown to induce an increase in proinflammatory cytokine production [165], while treatment with CBD reduced TNF-α production only in activated cells [156]. These results seem to suggest that CBD can act on activated cells without any effects on physiological cell homeostasis.

In animal models, CBD’s effects on cytokine production by PMNs are more contradictory; indeed, in a model in which rats were irradiated with ultraviolet light, Biernacki showed CBD to reduce TNF-α levels in PMNs, confirming the effect already assessed in ex vivo studies [164]. On the other hand, in a mouse model in which pulmonary inflammation was induced by LPS, CBD significantly enhanced the mRNA expression for several cytokines, such as TNF-α, IL-6, IL-17A, IL-23, and G-CSF [161], suggesting a proinflammatory role for CBD.

PMNs present different phenotypes, according to their stage of differentiation, age, and response to the surrounding microenvironment. Concerning the response to inflammatory conditions, PMNs can acquire different phenotypes, named N1 and N2, with pro- and anti-inflammatory functions, respectively [166]. Baban et al., using a mouse model of acute kidney injury, showed that CBD treatment induced an increase in N2 cells and a decrease in N1 cells, suggesting a prevalent anti-inflammatory effect for CBD in this model and opening the way to the consideration of novel uses of this safe compound in acute tissue injury [167]. Lastly, CBD inhibited the expression of COX-1 and COX-2 transcripts, but not prostaglandin PG-E2 production, in activated PMNs (Cosentino et al., 2022). 

Although with some exceptions, in general, CBD seems to exert a prevalent inhibitory effect on the proinflammatory functions of PMNs. In particular, CBD decreases cell adhesion to the endothelium and subsequent migration, as well as tissue infiltration and cell accumulation in inflamed tissues. Moreover, CBD seems to exert inhibitory effects on ROS and proinflammatory cytokine production, particularly in models in which PMNs were previously activated. This observation is of particular interest, because it suggests that CBD’s effects are restricted to inflamed tissues, thus slowing the inflammatory cascade without affecting physiological tissue homeostasis.

Overall, these findings allow suggesting a possibility for CBD employment in the treatment of diseases, such as rheumatoid arthritis, psoriasis, ankylosing spondylitis, Crohn’s disease, and other inflammatory bowel diseases [168,169,170,171], in association with specific drugs which have already demonstrated a therapeutic efficacy in these diseases.

The molecular targets of CBD for the modulation of PMN functions were studied by Thapa and colleagues, who found the CBD-induced reduction in PMN infiltration in the cornea was completely blocked by a 5-HT1A receptor antagonist but was still present after CB1 receptor antagonist treatment or in CB2R^−/−^ mice. The authors postulated that cannabinoid receptors are not involved in the infiltration mechanisms of PMNs, but the serotonin receptor seems to play a role in the process [158]. These results are in disagreement with those by Biernacki and colleagues, who indeed found that CBD treatment reduced CB1R expression, and they suggested that this receptor could be involved in ROS and TNF-α production [164].

## 3. CDB Modulation of Innate Immunity: Possible Implications for Disease Treatment

### 3.1. Atherosclerosis

Atherosclerosis is a dominant cause of cardiovascular pathologies, in which inflammation plays a crucial role. Typically, atherosclerosis occurs in the endothelium of large and medium arteries, where upregulation of adhesion molecules allows immune cells, such as mononuclear leukocytes (monocytes, B and T cells, dendritic cells, and mast cells) and PMNs to attach to the endothelium and cross into the intima [172]. Innate immune cells are abundant in atherosclerotic lesions [141,150,173,174] and produce proinflammatory cytokines which in turn increase local tissue inflammation.

It is well known that, during inflammation, CB2 receptors contribute to the recruitment of leukocytes by modulating chemotaxis, such as during myocardial ischaemia/reperfusion, and the disease progression can be inhibited, e.g., by THC in a CB2-dependent manner [175].

Altogether, these observations allow hypothesising that atherosclerosis could be treated by modulating the endocannabinoid system; therefore, CBD would also be recommended for this purpose, since it is a much safer drug than THC, and its potential preclinical and clinical use in the treatment of this widespread pathology is worth investigating.

### 3.2. Neurodegenerative Diseases

A number of incurable and debilitating conditions are associated with neurodegeneration, which is due to the progressive degeneration of neurons or neuronal apoptosis [176]. Neurodegeneration itself can occur in two different conditions; acute neurodegeneration is characterised by synaptic and axon degeneration, which can be the result of direct injury and damage to the neuron (e.g., in stroke, head trauma, cerebral or subarachnoid haemorrhage, and ischaemic brain injury resulting from foetal or perinatal hypoxia), whereas chronic neurodegeneration is instead characterised by aberrant signal transduction, excitotoxicity, synaptic dysfunction, oxidative stress, proteotoxicity and protein misfolding, excessive cell death, glia-supported neuroinflammation, and failure of neurogenesis (e.g., in Alzheimer’s disease). [177,178].

Furthermore, an autoimmune component has been hypothesised in neurodegenerative diseases such as Parkinson’s, in which peptides derived from α-synuclein, a crucial aggregate protein in Parkinson’s aetiology, act as antigens triggering cytotoxic and helper T-cell responses [179,180] and multiple sclerosis, in which autoimmunity appears to be mediated by myelin-autoreactive leukocytes such as T cells, B cells, and macrophages, which in turn cause neurodegeneration [181,182].

#### 3.2.1. Multiple Sclerosis

Innate immunity plays an important role in the aetiology and progression of multiple sclerosis [183].

In multiple sclerosis, the role of neuroinflammation is supported by abundant evidence, and approved disease-modifying therapies indeed rely on anti-inflammatory and immunomodulating effects [184]. 

CBD was shown to reduce macrophagic infiltration in the CNS in a murine model of EAE [103]; moreover, its described anti-inflammatory effects on glial cells could also provide an interesting opportunity for neuroprotection in multiple sclerosis [94,95,185]. For these reasons, CBD’s potential as a disease-modifying treatment is worthy of further investigation.

#### 3.2.2. Parkinson’s Disease

Parkinson’s disease is a progressive neurological pathology, characterised by a large number of motor and nonmotor symptoms, which impact physiologic functions [186]. In PD, neuroinflammation is a key point for disease progression; the main immune changes during PD occur in the brain and involve innate immunity, such as microglial cells [187]. However, alterations in the peripheral immune system have also been observed in PD patients [188,189,189,190,191,192,193,194], where the innate immune system is also found to be altered, as is the case for monocytes and the PMN-to-lymphocyte ratio [187,195].

CBD has already been proposed as a symptomatic treatment, which is usually prescribed when classical dopaminergic drugs fail to achieve adequate symptom control [18,196]. Although different immune-modulating approaches have already been tested, with some success in animal models, results of similar trials in human patients have so far proven less promising, with no disease-modifying treatment yet available for these patients [180]. The interesting anti-inflammatory and potentially neuroprotective properties of CBD in the disease are, therefore, worthy of further and deeper investigation [197].

#### 3.2.3. Alzheimer’s Disease

Alzheimer’s disease is the most common cause of progressive cognitive decline. Neuroinflammation has been hypothesised to be an important contributing factor to its pathogenesis and microglia, as resident immune effector cells of the CNS play a crucial role in regulating brain homeostasis and mediating innate immune responses in the disease [198,199]. Activated microglia commonly upregulates expression of pattern recognition receptors (PRRs), inducing an inflammatory response and secretion of proinflammatory cytokines, including IFNs [200,201]. 

The particularly safe profile of CBD also makes it a promising treatment in Alzheimer’s disease. Its action as a CB2R agonist could reduce inflammatory responses, with a consequent reduction in β-amyloid production and deposition, thus increasing cell viability and promoting glucose uptake in the brain. In AD models, this indeed resulted in cognitive improvement [202].

Furthermore, in a recent paper, Ribaudo and Landucci et al. also highlighted an interesting inhibiting effect by CBD on PDE9 [203], which is a promising mechanism of action against neurodegeneration [204,205].

### 3.3. Neuropsychiatric Diseases

Recent findings point towards the role of inflammation in several psychiatric diseases, such as schizophrenia [206,207,208] and bipolar disorder (BD) [209,210,211,212]. 

Over a century ago, schizophrenia was hypothesised to be associated with the immune system, and some studies indicated that both infections and inflammation might play a role in the disease. Moreover, neuroinflammation in schizophrenia seems to be characterised by the activation of microglial cells [213]. Furthermore, the implication of the immune system has also been documented in BD, where the levels of proinflammatory cytokines in affected patients were found to be increased in comparison with healthy controls [214,215].

Some antipsychotic drugs were also reported to restore a normal inflammatory profile in affected patients [212,216,217,218]. Recently, cannabinoids were also proposed as a possible treatment for psychoses, but only a few studies are available in this regard, and reported results are controversial. 

CBD-induced inhibition of astrocyte activity has been suggested as a possible treatment option in schizophrenia and other neuropsychiatric diseases [219,220], where neuroinflammation seems to be related to a downregulation of PPAR receptors and/or their endogenous ligands [221,222,223].

Whether the use of cannabinoids plays a protective or harmful role in the pathogenesis of these diseases remains unclear [224,225,226].

Promising results of anti-inflammatory treatments in decreasing BD symptoms, especially depressive ones, have been described [227,228]; thus, in this context, the emerging role of CBD in the modulation of inflammatory processes allows postulating that CBD could be useful, in association with standard therapies, for the treatment of neuropsychiatric diseases in which an inflammatory component is involved in disease progression [208] and/or therapeutic response [212,216,217,218].

### 3.4. Autoimmune/Inflammatory Diseases

Crohn’s disease is a chronic inflammatory intestinal disease. The evolution of Crohn’s disease is based on tissue inflammation caused by a strong immune response against luminal bacterial antigens; immune cells such as T cells, monocytes, and NKs are involved in this process as they infiltrate the gut of patients [229].

Osteoarthritis (OA) is the most common joint disease in the world, with an age-associated increase in both incidence and prevalence [230]. Evidence describes a pivotal role for both macrophages and the complement system in joint inflammation and disease progression, pointing towards a strong correlation between OA and the innate immune system [231].

Furthermore, in several rheumatic diseases (rheumatoid arthritis, systemic lupus erythematosus, and ANCA-associated vasculitis), complement activation is suggested to play a crucial role [232]. Moreover, the IL-1 family of cytokines is associated with innate immune responses, which characterise rheumatic diseases [233].

Despite CBD already being used in the treatment of different autoimmune and inflammatory conditions, such as Crohn’s disease [234], osteoarthritis [235], and rheumatic diseases [236,237], predominantly preclinical data support the hypothesis that anti-inflammatory effects of the compound are directly involved in its clinical effects [19,20,21].

### 3.5. Other Therapeutic Possibilities

The effects of CBD on the modulation of immune responses are interestingly being investigated in various other diseases, where targeting immune mechanisms might provide viable treatment options.

Recently, a clinical protocol was designed to evaluate whether CBD could act as a neuroinflammation inhibitor in patients with chronic lower-back pain, by affecting glial activation and reducing levels of translocator protein binding. The clinical trial is ongoing [238].

Viral diseases are another field in which treatment with CBD has shown promising preclinical results, which should be further investigated in appropriate clinical trials [239].

In cancer [240], asthma [241], and cardiovascular diseases, inflammation also plays an important pathophysiologic role. Thus, CBD should be tested in these contexts, where its particularly safe profile will allow for a seamless addition to standard therapies.

## 4. Conclusions

Altogether, these observations suggest future potential applications, particularly in chronic neuroinflammation. Despite wide heterogeneity in the approaches used for the evaluation of CBD actions on innate immune functions, the examined studies showed a predominant inhibitory effect of CBD on inflammatory processes. Interestingly, the effects of CBD are quite different in different cell subpopulations involved in innate immunity. For example, CBD exerts a prevalent inhibitory effect on astrocytes [92,93,105] and PMN [26,36,94,153,155,156,157,158,159,160,164], whereas, in microglia, oligodendrocytes, monocytes/macrophages, and mast cells, results are contradictory, with some authors reporting anti-inflammatory effects for CBD [90,90,99,101,102,103,104,105,106,110,111,116,117,118,119,120,121,122,124,125,126,127,129] and other authors reporting proinflammatory effects [89,107,108,110,111,120,122,123,124,125,129]. Lastly, to the best of our knowledge, no studies have reported CBD’s effects on dendritic cells, NK cells, eosinophils, and basophils.

Due to the safe profile of CBD and its already widespread use in clinical settings, clinical trials are all the more important to test the clinical efficacy of this compound in several diseases, which are now lacking in both symptomatic and disease-modifying therapies.

However, despite the possible exceptional value of CBD’s anti-inflammatory effects in clinical practice, CBD-based drugs are so far prescribed only as comedication in subjects with poor clinical response, or in which standard therapies induce intolerable side-effects. Quite surprisingly, we could not find any published clinical study which was specifically designed to evaluate the anti-inflammatory role of CBD treatment. In this context, results reported in this review could support a more rational use of CBD in diseases with a strong inflammatory component.

## Figures and Tables

**Figure 1 ijms-24-03125-f001:**
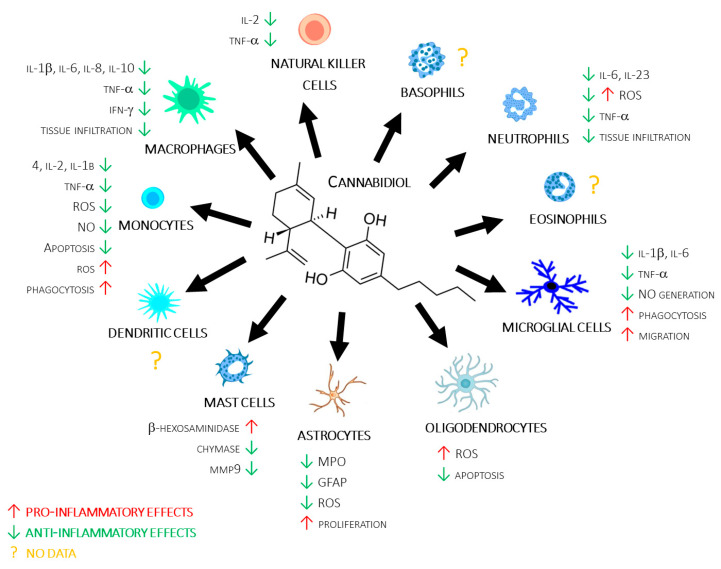
CBD’s molecular structure, along with a summary of its reviewed actions on different immune cells.

**Table 1 ijms-24-03125-t001:** Cannabinoid-based drugs.

Cannabinoid Components	Brand Name	Origin	Indication(s)	Authorisation
Nabiximols (Δ9-THC and CBD 1:1)	Sativex	Natural	Multiple sclerosis-related spasticity and neuropathic pain, and cancer-related pain	FDA/EMA
Nabilone (Δ9-THC analogue)	Cesamet/Canemes	Synthetic	Chemotherapy-induced nausea and vomiting	FDA only
Dronabinol ((-)-trans-Δ9-THC)	Marinol/Syndros	Synthetic	Anorexia related to AIDS, and nausea and vomiting induced by chemotherapy	FDA only
CBD	Epidyolex	Natural	Epilepsy	FDA/EMA
CBDV	GWP42006	Natural	Rett syndrome, fragile X syndrome *, and autism **	/
(Whole plant)	Cannabis FM2, Cannabis Flos, Pedanios	Natural	Neuropathic and spasticity-associated pain, nausea and vomiting due to chemo/radiotherapy and HIV therapy, anorexia and cachexia, glaucoma, and Tourette syndrome	AIFA

Δ9-THC = Δ9-tetrahydrocannabinol; CBD = cannabidiol; CBDV = cannabidivarin. * Received FDA/EMA orphan designation; ** ongoing trial.

**Table 2 ijms-24-03125-t002:** CBD pharmacodynamics.

Target	Action	Affinity (pKi)	Reference(s)
CB1	Weak agonist, negative allosteric modulator	3.64	[38,39]
CB2	Weak agonist, inverse agonist	3.46	[38,39]
TRPV1, TRPV2, TRPV3, TRPA1	Agonist	N/A	[33]
TRPM8	Antagonist	N/A	[33]
PPARγ	Agonist	N/A	[40]
GPR55	Antagonist	N/A	[41]
GPR3, GPR6, GPR12	Inverse agonist	N/A	[41]
5-HT1A, 5-HT2A, 5-HT2C	Agonist	1.1 (5-HT2C)	[35,39,42]
A2A	Agonist	N/A	[36]
μ, κ, and δ opioid receptors	Allosteric modulator	1.3 (µ), 2.3 (κ), 6.4 (δ)	[37,39]
GLRA1, GLRB	Allosteric modulator	N/A	[42]
GLRA3	Potentiator	N/A	[42]
GPR18	Unknown	N/A	[42]
ADRA2B, ADRA2C	Unknown	3.2 (B), 3.7 (C)	[39]
PDE9	Inhibitor	N/A	[43]

CB1 and CB2 = cannabinoid receptors type 1 and 2; TRPV1, TRPV2, and TRPV3 = transient receptor potential cation channel subfamily V (vanilloid) members 1, 2, and 3; TRPA1 = transient receptor potential cation channel A1; TRPM8 = transient receptor potential cation channel subfamily M (melastatin) member 8; PPARγ = peroxisome proliferator-activated receptor γ; GPR55, GPR3, GPR6, and GPR12 = G protein-coupled receptors 55, 3, 6, and 12; 5-HT1A, 5-HT2A, and 5-HT2C = serotonin receptors 1A, 2A, and 2C; A2A = human adenosine A2A receptor; GLRA1, GLRB, and GLRA3 = glycine receptor subunits alpha-1, beta, and alpha-3; GPR18 = G protein-coupled receptor 18; ADRA2B and ADRA2C = α2B and α2C adrenergic receptors; PDE9 = phosphodiesterase 9.

**Table 3 ijms-24-03125-t003:** CBD pharmacokinetics.

Characteristic	Value	Note	Reference(s)
Oral bioavailability (fasting)	~6%	Greatly increases with food intake (about 4-fold increase for both C_MAX_ and AUC)	[44]
T_MAX_	2.5–5 h	/	[45]
T_1/2_	10–17 h	/	[45]
Binding to plasma proteins (%)	≥88%	/	[45]
Vd	20,963–42,849 L	/	[45]
CL	1111–1909 L/h	Mainly hepatic clearance	[45]

**Table 4 ijms-24-03125-t004:** SNPs with potential effects on CBD pharmacology.

Protein	Gene	Variant	Nucleotide Change	AF (%)	Biological Effect
Cannabinoid receptor 1 (CB1R)	*CNR1*	rs806368	T>A	21	Associated with alcohol dependence [51]
		rs806380	A>G	34	Associated with alcohol dependence [51]
		rs1049353	C>A	27	Associated with alcohol dependence [51]
		rs2023239	T>C	17	Increased cannabinoids-induced side-effects [51]
Cannabinoid receptor 2 (CB2R)	*CNR2*	rs2229579	G>A	10	Increased mRNA expression [52]
Transient receptor potential cation channel subfamily V member 1 (TRPV1)	*TRPV1*	rs8065080	T>C	38	Associated with hypoalgesia [53,54]
		rs222747	C>A	75	Associated with hypoalgesia [53,54]
		rs4790521	T>C	32	Association with the COPD risk [55]
ATP-binding cassette subfamily C member 5 (ABCC5)	*ABCC5*	rs3749442	G>A	17	Increase in CBD response [47]
P-glycoprotein (P-gp)	*ABCB1*	rs2032582	A>C	55	Reduced P-gp expression and activity [56]
		rs1045642	A>C	48	Reduced P-gp expression and activity [56]
		rs1128503	A>G	57	Reduced P-gp expression and activity [56]
Solute carrier family 15 member 1 (SLC15A1)	*SLC15A1*	rs1339067	A>C	66	Decreased transporter expression in CNS and lower response to CBD [47]
Cytochrome P450 2C9 (CYP2C9)	*CYP2C9*	rs1799853 (*2)	C>T	12	Decreased enzyme activity [57]
		rs1057910 (*3)	A>C	7	Decreased enzyme activity [57]
Cytochrome P450 3A4 (CYP3A4)	*CYP3A4*	rs35599367	G>A	5	Decreased enzyme activity [58]
Aldehyde oxidase 1 (AOX1)	*AOX1*	rs6729738	C>A	53	CBD low seizure control in epilepsy [47]
Amine oxidase copper 1 (AOC1)	*AOC1*	rs12539	C>T	22	Increased diarrhoea and low seizure control in epileptic patients [47]
UGT1A9 (UDP-glucuronosyltransferase 1A9)	*UGT1A9*	rs72551330 (*3)	T>A	2-3	Decreased enzymatic activity [59]
		rs3832043 (*22)	(DT) 9>10	40	Increased enzymatic activity [60]

CBD = cannabidiol; COPD = chronic obstructive pulmonary disease; LFT = liver function test; HTR3E = 5-hydroxytryptamine receptor 3E; NR3C1 = nuclear receptor subfamily 3 group C member 1; AF = allelic frequencies in Caucasian population.

## Data Availability

Not applicable.

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
