# Peer review of "Effects of Cannabidiol on Innate Immunity: Experimental Evidence and Clinical Relevance"

_ijms, 2023, doi:10.3390/ijms24043125_

Round 1

Reviewer 1 Report

The manuscript from Martin et al. discusses how Cannabidiol affects cells of Innate Immunity and provide the importance of this immunomodulatory potential for treatment of inflammation-related diseases.

The authors collected the available information for each cell type. Unfortunately, for some cells the data are scarce. After this, the authors provide the examples of diseases where CBD have shown therapeutic effects in pre-clinical trials, 

The manuscript is well written and organized, however, the authors could make the paragraphs size more uniform. Please note that some paragraphs only have 2-3 lines while others are longer. 

The authors could also include a figure with the structure of CDB. Other figures illustrating the effects of CDB are also welcomed.

It is also necessary to provide more details about the models used in each assay. For instance, the authors say somewhere that the study used mice but do not explain what kind of mice, sex, age… This information is important. The same for in vitro or ex vivo models, please add more information where is necessary.

Other comments are provided in the attached pdf file.

Author Response

We thank the reviewer for his positive and constructive comments, which allowed us to further improve our manuscript.

Following the reviewer's suggestion, the following changes were made to the main manuscript and the supplementary material:

  • Line 11: the name of the plant was made italic
  • Line 17: sentence was edited by adding experimental models in which mentioned pieces of evidence were found: "[...]in different models, including mice, rats, guinea pigs and even ex vivo experiments on cells from human healthy subjects[...]"
  • Line 19: some main mechanisms of action of CBD were added in the abstract.
  • Line 28: corrected "Cannabis sativa L."
  • Line 32: phrase was split in two separate sentences and edited for clarity.
  • Line 165: inserted Figure 1, which summarises the reviewed CBD actions on different immune cells.
  • Line 353-354: edited paragraph title and the first sentence to better define PMNs as polymorphonuclear neutrophils.
  • Sections 2.2 through 2.7: paragraphs order was changed to better emphasise relevant information and improve the flow of the manuscript.
  • Table S1: detailed information on the listed animal and other experimental models was added wherever it was missing and available in the source material.

Reviewer 2 Report

The information gathered is relevant to the area of ​​cannabinoids, an emerging area. The authors did an excellent job. The article is well-structured and organized. The presented data is properly presented and interpreted.

Line 11: The name of the plant must be in italic. Please revise the manuscript.

Author Response

We thank the reviewer for his kind comments on our work. We corrected the manuscript following his and other reviewers' advice.

Round 2

Reviewer 1 Report

The authors have improved the manuscript following the suggestions. The manuscript is now suitable for publication in this journal.